# Crusade: The Arising of a Concept Based on Portuguese Written Records of Three Military Campaigns (1147–1217)

**Paula Pinto Costa** [1,*] and **Joana Lencart** [2,*]

1   Department of History, Political and International Studies, and Transdisciplinary Culture,
    Space and Memory Research Centre, University of Porto, 4150-564 Porto, Portugal
2   Transdisciplinary Culture, Space and Memory Research Centre, University of Porto, 4150-564 Porto, Portugal
*   Correspondence: ppinto@letras.up.pt (P.P.C.); jlencart@letras.up.pt (J.L.)

**Abstract:** The historiography of the crusade and reconquest in the Iberian Peninsula, and Portugal in particular, dates from the beginning of the twentieth century. Since the 1920s, it has been assumed that the reconquest was an output of the crusade in the Iberian West due to the so-called "Bull of Crusade" given to Portugal. The "idea of the Crusade" in Portugal was enhanced by Carl Erdmann in the 1930s and 1940s. This interpretation has been endorsed by the very context in which the Kingdom of Portugal emerged and developed throughout the twelfth and thirteenth centuries. From the 1980s onwards, the launching of systematic research on the military orders also reinforced this perspective. The deep affinity between the military orders and the crusades in the context of the reconquest is reflected in this historiography. These concepts—military orders, crusade, reconquest—have been studied without distinction being made between them, adding to the complexity of this analysis. Considering the historiographical achievements regarding the crusade, it is pertinent to rethink the associations between reconquest, crusade and military orders. Reading certain historical narratives is crucial for this analysis, although the written records do not fully deplete the subject. To reinforce the relevance of this approach, we will also consider royal and pontifical diplomas. Tracing the terminology used in these documents and identifying how these historical realities were referred to are the two main goals of this paper. For that purpose, three key moments of the Portuguese reconquest have been chosen: the conquests of Lisbon (1147), Silves (1189), and Alcácer do Sal (1217). These have one feature in common: the presence of crusaders travelling to the Holy Land, which supports the terminological analysis of those discourses. Different perspectives are embodied in these conquest narratives when compared with royal and papal diplomas on the same issue and of a similar chronology. Historiography on Reconquest, crusade, and the military orders is often conditioned by ideology, occasionally revealing a tendency to repeat ideas without debating them. This paper's analysis is based on the aforementioned written records and is undertaken in order to verify when the word crusade/crusader appeared in Portugal, to assess to what extent the war of conquest in Portuguese territory followed the example of the holy war and to evaluate the commitment of the crown and the Holy See in this complex process.

**Keywords:** crusade; *reconquest*; military orders; *cruce signati*; the Middle Ages

## 1. Introduction

The study of the crusade and reconquest in the Iberian Peninsula, and Portugal in particular, has been the subject of numerous historiographical contributions. The first studies on the origins and establishment of the so-called "Bull of Crusade" in Portugal date back to the 1920s, and historians accept that the Iberian Reconquest was a product of the crusade in the Iberian west (Caldas 1923). The historiography used the expression "Bull of Crusade" to report the spiritual and financial papal support to the military expeditions against the enemies of the faith (Weber 2011, p. 15; Goñi Gatzambide 1958). Carl Erdmann, in the 1930s and 1940s, emphasised this association by assuming that there was an "idea

of Crusade" in Portugal (Erdmann 1935, 1940). This view is supported by the historical context of the formation of the Portuguese Kingdom. From the 1980s, this perspective has been reinforced by systematic research on the military orders in Portuguese academia. Although they were different from one another as institutions, they had a great affinity with the crusade, something which is reflected in the historiographical outputs. Since then, the historical approach has become increasingly complex and confusing, as the reconquest, crusade and the military orders intersect. Considering the most recent historiographical achievements on how to characterise these concepts, it is prudent to rethink such associations. Concerning the concept of the crusade, many contributions have been brought to the discussion, and the approaches to crusade narratives are of much interest to historians (Markowski 1984; Cosgrove 2010; Ayala Martínez 2013; Ayala Martínez et al. 2016; Maier 2021; Weber 2021; Buck and Smith 2022). The meaning of the word crusade has changed over time, and only in the seventeenth century did the word itself become used to tell the history of the crusades (Weber 2011; Maier 2021). Until this century, other words were used to refer to this concept. The existing scholarship justifies the relevance of this approach and is based on linguistical reflections on the military campaigns which took place in Portugal between 1147–1217 that were generally connected with the crusade environment. Accepting that contemporaries did not use the word crusade, it is important to identify the terminology they used so that we may carry out a deep discussion of the concept of the crusade (Weber 2021, p. 199), and to improve the systematic research on this issue (Maier 2021, p. 15).

According to C. Erdmann, between 1140 and 1217, eight fleets of crusaders from northern Europe were persuaded to fight the Muslims in Portugal (Erdmann 1940, p. 16). In an article for the *Dicionário de História Portuguesa*, Ruy d'Abreu Torres wrote about the "Cruzados na conquista de Portugal" (Crusaders in the conquest of Portugal), reporting on the support provided by these northern-European knights, who were generally called crusaders, to the Portuguese Reconquest, from 1140, when King Afonso I first attempted the conquest of Lisbon, until 1217, when Alcácer do Sal was also conquered (Torres 1985, pp. 247–50). Due to their intervention, the sieges of Lisbon (1147), Silves (1189) and Alcácer do Sal (1217) were successful. Lisbon was conquered in the context of the Second Crusade, Silves in the Third Crusade and Alcácer do Sal in the Fifth Crusade. These military campaigns reinforced the connection between the crusaders and the Portuguese Reconquest.

Although conquest narratives are crucial to this analysis, this type of written source does not exhaust the study of the terminology used to report such events. Royal documents, as well as the papal diplomas, contain linguistic resources that reinforce the relevance of this approach. Thence, the goal is to survey the terminology used in these three document groups and to identify how such historical events were designated. To identify when the word crusade began to be used, it is necessary to understand the context in which it was applied. Taking into account the three historical events already pointed out (dated 1147, 1189 and 1217), we have selected certain words, expressions and discursive features used to report and describe these episodes. The aim of this methodological exercise is to bring out the singularities of each case and to comprehend them in the light of the bibliography available on this topic.

## 2. Written Records

The written records selected are organised into three groups: contemporary narratives of the conquest of Lisbon (1147), Silves (1189) and Alcácer do Sal (1217); royal chancery records; and papal diplomas. For this terminological analysis, only original manuscripts—contemporary to the reported facts—were examined, to avoid any textual variations that could have been introduced in later copies. This methodology makes it possible to identify the impact of these three military campaigns on contemporary literary memoirs and royal and papal chancery belonging to the same period.

The narrative of the conquest of Lisbon—*De Expugnatione Lixbonensi*—describes in detail the crusading contingent's passage through Portugal and its role, at King Afonso Henriques' request, in the siege and capture of the city in October 1147. This account of the conquest is a letter written between the second half of the twelfth century and the first decade of the following (De Expugnatione Lyxbonensi. A conquista de Lisboa: Relato de um Cruzado 2001). Its author is a Norman man who embarked on the great enterprise of the Second Crusade, reporting to Osbert of Bawdsey on the successes and progress of the crusaders on their journey to the Holy Land (Branco 2001, pp. 9–12). This text was first published in *Portugaliae Monumenta Historica* in 1861 (Portugaliae Monumenta Historica a saeculo octavo post Christum usque ad quintumdecimum iussu Academiae Scientiarum Olisiponensis edita. Scriptores 1861, pp. 391–405) and, as Maria João Branco has summarised, has since been republished, in Portugal and elsewhere, in numerous editions (Branco 2001, pp. 9–51).

The narration of the siege and conquest of Silves, a small town in the Al-Gharb (Algarve), in the far south of Portugal, can be found in *Narratio de Itinere Navali Peregrinorum Hierosolymam Tendentium et Silviam Capientium, A. D. 1189*, a text edited by Charles Wendell David in 1939 (Narratio de Itinere Navali Peregrinorum Hierosolymam Tendentium et Silviam Capientium, A. D. 1189 1939). This text resembles a travel itinerary, in which the events experienced by the crusaders on their journey to the Holy Land as part of the Third Crusade in 1189 (Azevedo 2012, pp. 9–10) are recounted in detail. Its author, probably a pilgrim of Germanic origin, wrote at the end of the twelfth century. The narrative ends in Silves, so we do not know whether or not these pilgrims reached the Holy Land, their original destination (Azevedo 2012, p. 10).

The account of the siege and conquest of Alcácer do Sal, or simply, Alcácer, is known by the so-called *Gosuini de expugnatione Salaciae carmen*, after the name of its author. Indeed, Gosuíno called it a "poem", giving it a moral rather than a literary or historical purpose (Nascimento 2012, p. 494). The manuscript of this chronicle can be found in the Portuguese National Library (Biblioteca Nacional de Portugal, ALC. 415, fls. 149v-151r) and it is dated from the early thirteenth century. A Portuguese translation was first published in 1632 (Poem by Gosuíno, in Monarquia Lusitana, Parte Quarta 1974, pp. 133–6) and a bilingual edition (Latin and Portuguese) was published in 2005 by Aires do Nascimento (Nascimento 2005, pp. 619–63), and re-edited in 2012 (Nascimento 2012, pp. 503–16). Recently, in 2021, another bilingual edition was published, in Latin and English, by Jonathan Wilson (Wilson 2021). Maria Teresa Pereira also published an article on this chronicle, providing a detailed approach (Pereira 1996, pp. 321–57).

The definitive conquest of Alcácer do Sal was achieved in 1217 after a siege on the initiative of Soeiro Viegas, bishop of Lisbon. It included the participation, among other military forces, of Nordic crusaders on their way to Palestine (as part of the Fifth Crusade) and of the military orders, namely those of the Temple, Hospital and Santiago (Martins 2007, p. 199, nt. 888 e p. 607, nt. 225).

Other contemporary documents can be examined to complete the analysis. Among these, the diplomas issued by the papacy and the royal chancery have particular relevance, as the main sponsors of those military campaigns enacted them. On the one hand, the papacy wished to expand Christendom; and on the other, the crown sought to conquer new lands, pushing the Peninsula's southwestern frontier.

## 3. Terminology and Discursive Resources

Having identified and briefly presented the three typologies of historical records, we may select the most paradigmatic expressions and discursive resources about the historical episodes which literature from the twentieth century refers generally to as crusade and reconquest.

### 3.1. Contemporary Narratives of Conquest

Taking into account the main strand of this study, we will first analyse the terminology used in the narrative of the conquest of Lisbon—*De Expugnatione Lyxbonensi*—following Aires Nascimento's transcription (De Expugnatione Lyxbonensi. A conquista de Lisboa: Relato de um Cruzado 2001). Despite the long narrative, references to Muslims, or the enemy, are scarce. The crusading fleet's first stop in Portuguese territory was in Porto, where the northmen were received by the local bishop, Pedro Pitões. In an energetic sermon, the bishop lamented the occupation of the peninsula by the Muslims, something he classified as a punishment from God: "*Auditum satis partibus vestris credimus, quod divina ultio superincumbentibus Mauris et Moabitis totam Hyspaniam in ore gladii percusserit*" (De Expugnatione Lyxbonensi. A conquista de Lisboa: Relato de um Cruzado 2001, p. 66). The bishop encouraged them to stay in Iberia. In his perspective, the worthy purpose is not to have been to Jerusalem but to have lived a good life while on the way (Tyerman 1995, p. 563).

Muslims are mentioned again, as those who have come to disturb the inhabitants of the peninsula: "*A Mauris enim semper inquietatis*" (De Expugnatione Lyxbonensi. A conquista de Lisboa: Relato de um Cruzado 2001, p. 82). In other parts of the text, the author mentions the "other" as "*adversarii crucis*" (De Expugnatione Lyxbonensi. A conquista de Lisboa: Relato de um Cruzado 2001, p. 70). In short, he considered that the Moors and the Moabites were wrongfully occupying the peninsula. On the eve of the final attack, the bishop of Lisbon delivered a sermon holding the relic of the True Cross in his hands, referring to the Moors as "*omnium impurissimi nobis calumpniantur*" (De Expugnatione Lyxbonensi. A conquista de Lisboa: Relato de um Cruzado 2001, p. 116 e p. 120). By contrast, the people committed to this military purpose were identified as members of "*peregrinationis*" (De Expugnatione Lyxbonensi. A conquista de Lisboa: Relato de um Cruzado 2001, p. 56); "*felici peregrinatione comutasse*" (De Expugnatione Lyxbonensi. A conquista de Lisboa: Relato de um Cruzado 2001, p. 62); "*servi crucis*" (De Expugnatione Lyxbonensi. A conquista de Lisboa: Relato de um Cruzado 2001, p. 70); or "*boni milites*" (De Expugnatione Lyxbonensi. A conquista de Lisboa: Relato de um Cruzado 2001, p. 70).

Although these people were motivated to participate in the battle and stay in Portugal, they preferred to continue their journey to the Latin East. According to the text, their only expectation was to profit from the looting, and they were accused of having non-charitable goals: "*certe peregrinatio vestra non videtur karitate fundata*" (De Expugnatione Lyxbonensi. A conquista de Lisboa: Relato de um Cruzado 2001, pp. 87–88). Although they were pilgrims, the economic intentions were indeed decisive in their actions. For instance, after the Muslim surrender of Lisbon, the pilgrims who attended the siege took part in a riot for the looting, and they did not respect the Portuguese King's instructions not to enter the city (De Expugnatione Lyxbonensi. A conquista de Lisboa: Relato de um Cruzado 2001, p. 138).

About 40 years after the conquest of Lisbon, a fleet from northern Europe, on its way to the Latin East, was called upon to intervene in the conquest of the southern Portuguese coast. An additional effort was needed to conquer Silves, which would be reached precisely at a time of the king's taking initiative in warfare (Branco 2006, pp. 87–102). We will focus on the terminology used by the author of "*Narratio de Itinere Navali Peregrinorum Hierosolymam Tendentium et Silviam Capientium, A. D. 1189*" to refer to the "other", using expressions such as enemies, Saracens, and Moors: "*Nostri ergo per inimicorum terras nimis avide et incaute discurrerunt*" (Narratio de Itinere Navali Peregrinorum Hierosolymam Tendentium et Silviam Capientium, A. D. 1189 1939, p. 618); "*Maurus*" and "*Sarraceni*" (Narratio de Itinere Navali Peregrinorum Hierosolymam Tendentium et Silviam Capientium, A. D. 1189 1939, p. 624). The name by which they were known varied according to their place of origin: "*Sarraceni autem in Ispania habitantes Andeluci dicuntur, qui in Africa, Mucimiti vel Moedini, qui in Marorlce, Moravidi*" (Narratio de Itinere Navali Peregrinorum Hierosolymam Tendentium et Silviam Capientium, A. D. 1189 1939, p. 621). Besides these designations, the author employs adjectives to qualify them in a derogatory way: "*Sarracenis agressis*" (Narratio de Itinere Navali Peregrinorum Hierosolymam Tendentium et Silviam Capientium, A. D. 1189 1939, p. 625).

In this chronology, there are no terminological references to crusaders as such. The author associates the individuals from northern Europe to pilgrims, considering their journey to be a pilgrimage (Chléirigh 2014): "*faciendo [VIII] dies peregrimus*" (Narratio de Itinere Navali Peregrinorum Hierosolymam Tendentium et Silviam Capientium, A. D. 1189 1939, p. 615). He also points out that the city of Lisbon had been taken with the help of their counterparts, i.e., pilgrims, some decades before: "*Hec Ulixbona, opulenta et magna valde, ante quadraginta et IIIIᵒʳ annos a peregrinis nostris capta*" (Narratio de Itinere Navali Peregrinorum Hierosolymam Tendentium et Silviam Capientium, A. D. 1189 1939, p. 616). There is even a particular mention of pilgrims from the British Isles: "*peregrinorum de Britannia venit ad nos*" (Narratio de Itinere Navali Peregrinorum Hierosolymam Tendentium et Silviam Capientium, A. D. 1189 1939, p. 618). In fact, the Latin word "*peregrinus*" can be translated as one who comes from outside, a foreigner. Indeed, the pilgrim does not travel to holy places necessarily only for religious purposes (Thesavrvs lingvae latinae editvs ivssv et avctoritate consilii ab academiis societatibvsqve diversarvm nationvm electi 1982–1997, pp. 1307–15).

The author also mentions the presence of the military orders, at the king's side, during the siege and conquest of Silves: "*erant cum eo milites religiosi de tribus sectis [ … ] Item milites de ordine Cisterciensi [ … ] quorum caput est Callatravia in regno Castelle et Ebora in regno Portugalensi, sed Callatravia mater est et Ebora filia. Item Iherosolimitarum alii erant de Templo, alii de sancto Sepulcro, alii de Hospitali*" (Narratio de Itinere Navali Peregrinorum Hierosolymam Tendentium et Silviam Capientium, A. D. 1189 1939, pp. 630–31).

To sum up, the words "*Maurus*" and "*Sarraceni*" acquire a pejorative significance: they were regarded as aggressors. The men in northern European fleets who had fought in the siege were seen as "*peregrini*" travelling to the Holy Land. For this author, alongside the king were also the Jerusalemite Orders—Templars, Hospitallers and Holy Sepulchre—besides that of Calatrava, known in Portugal as "de Évora".

Considering the case of Alcácer do Sal, and according to the title of this narrative, its conquest was due to the Franks, "*Quomodo capta fuit Alcaser a Francis*" (Nascimento 2012, p. 503), an expression with a broad meaning. According to Aires Nascimento, the author, Gosuíno, could himself be a native of Flanders (Nascimento 2012, p. 502).

Interestingly, the poem's author suggests that these foreigners were not enthusiastic about the campaign: "*non hanc sed gentem nos debellare iubemur que Christi tumbam que loca sacra tenet*" (Nascimento 2012, p. 506). In comparative terms, the attraction that the Holy Land held when compared with the western territories, namely Iberia, was obvious. For these individuals, contributing to the Iberian cause could be frustrating. However, the poet does not attribute any particular motivation to them, as he associates them with executing a mandate, an order, as if they were mercenaries. To convince people to participate in this type of battle, some opinion-makers used to compare the war in the eastern Mediterranean to the war in the western Iberian peninsula, arguing that all were necessary and equally respected and prestigious (Weber 2016, p. 227).

For about a century, political and religious goals were being developed and expressed in the reinforcement of the pilgrimage routes towards Santiago de Compostela. Since 1120 (Marques 1999, pp. 177–214), Compostela was distinguished with the privilege of being an Apostolic See, which assured it a similar status as Jerusalem and Rome. However, it is debatable how much knowledge northern travellers had of this prerogative and how this might have encouraged their willingness to become personally involved in the Iberian war.

To assess the motivation of these individuals, it is crucial to take a critical approach to the vows they would take, a point also raised in the text: "*Vota peregrinos cupientes soluere quosdam*" (Nascimento 2012, p. 503). It should be emphasised that only some of those "*peregrini*" were willing to respect these vows. The use of the word "*peregrini*" in this context, associated with vows, raises further questions, insofar as only the crusaders took vows before starting the campaign. As already mentioned, the word "*peregrinus*" can also refer to the foreigner. The expression "pilgrims" is used by the poet to refer to the distribution of loot: "*Quotquod erant hostes et eorum res, peregrinis sedunt, pars inde quilibet*

*equa datur*" (Nascimento 2012, pp. 507–8). Only some pilgrims would be predisposed to respect their vows. In a document of 1217, to which we will refer further below, they were gathered in an "*exercitus peregrinorum*" (Monumenta Henricina 1960, vol. 1, doc. 25, p. 47). Thus, the pilgrim was compared with a crusader with a specific vow and not just to someone guided by devotion. The crusade would then be both a pilgrimage towards the Holy Sepulchre in Jerusalem and a war for its liberation. The documents written by the crusaders contain various terms to refer to pilgrimage, such as "*peregrinatione*", "*via*", "*iter*", "*iter beatum*", "*iter Domini*", "*sanctum iter*", with them being referred to occasionally as "*peregrini*", and initially, though less frequently, as serving under an "*exercitus*" (Riley-Smith 1997, pp. 155–66).

Returning to the poem on the conquest of Alcácer, the exhortation is addressed to the servers of Christ, considering them to be enemies of the foes of the Cross: "*O fratres, famuli Christi, crucis hostibus hostes*" (Nascimento 2012, p. 504). The intensification of the expression *enemy* suggests a need to legitimise the sense of hostility associated with these actions.

The conquest, according to this account, was attributed to the "*cruce signati*" (Nascimento 2012, p. 503). This expression should be underlined by its simultaneous use in other literature. The "*cruce signatorum*" and "*quedam navigii multitudo crucesignatorum de Alamannia et de Flandria et de aliis partibus*" (Monumenta Henricina 1960, vol. 1, doc. 25, p. 46) are mentioned again in a 1217 document addressed to Pope Honorius III, enacted by the bishops of Lisbon and Évora, the Master of the Order of the Temple in Spain, the Prior of the Hospital in Portugal, and the Commander of the Order of Santiago in Palmela. The authors of this missive reported the capture of Alcácer to the Saracens as "*Yspania expugnantibus sarracenos*" (Monumenta Henricina 1960, vol. 1, doc. 25, p. 46). In the same letter, the bishops and masters asked the pope for permission for the crusaders to stay in the peninsula for another year, to liberate it from the Moors. They thus asked for the indulgences of the Holy Land for that campaign as well as other requests, equating the struggle in the peninsula to that of the Holy Land. The linguistic and ideological resources of this document are more elaborate and emphasise certain value judgements: "*ad Yspaniam liberandam et ad inimicos sancte fidei expugnandum*"; "*et dictus exercitus peregrinorum, ad cultum perfidie paganorum de tota Yspania penitus extirpandum [ . . . ] et quod tam ipsi quam nostrates crucesignati*" (Monumenta Henricina 1960, vol. 1, doc. 25, p. 47).

At approximately the same time, William, a Dutch Earl and a constable of the foreign crusaders who had taken part in the siege of Alcácer, wrote to Pope Honorius III, recounting the Alcácer episode: "*in obsidione castri cuiusdam sarracenorum Alchazar, christiani valde inimicum et dampnosum*" (Monumenta Henricina 1960, vol. 1, doc. 26, p. 49). He tells how, with 100 vessels, he took part in the capture of that town from the Saracens, imprisoning 2000 of them, including Abur, the military leader responsible for the castle, who was baptised along with other 100 captives. He claimed that he should continue the campaign in the peninsula to free it from Islam, as desired by the Christian kings: "*Yspania sarracenorum in magna parte fidei catholice debeant subiacere*" (Monumenta Henricina 1960, vol. 1, doc. 26, p. 49). He also asked for instructions on whether the crusaders should remain in the peninsula or should head to the Holy Land (Monumenta Henricina 1960, vol. 1, doc. 26, pp. 48–49).

### 3.2. Pontifical Written Records

To enrich our critical reflection on the narrative sources, the documents produced by the pontifical curia hold special interest. Generally, these records reveal a refinement of the discourse as time progressed. Earlier references to Muslims are scarce, even when they might be expected, considering their settlements in the Iberia. For example, in Adrian IV's bull to the Templar master and friars in 1159 on the construction of churches in Ceras, a territory that included Tomar (the place that was to become the headquarters of the Order of Christ), there is no reference to Muslims (Monumenta Henricina 1960, vol. 1, doc. 5, pp. 12–13).

After the conquest of Silves, Pope Clement III enacted two bulls containing discursive resources significantly different from those used in the so-called crusader narratives and the terminology used in royal documents. In the "*Manifestis probatum*" bull, of 7 May 1190, the pope congratulated the king for the services rendered to the Church through the victories over the enemies of the Catholic faith. Given that this papal diploma is the confirmation of the earlier "*Manifestis probatum*", of 23 May 1179, by which the Holy See legitimised the King of Portugal, it is evident that this diploma reproduces the discourse already conveyed by the previous one. In both, there are expressions such as "*inimicorum christiani nominis intrepidus extirpator*", and "*omnia loca que cum auxilio celestis gratie de sarracenorum manibus eripueris*" (Monumenta Henricina 1960, vol. 1, doc. 9, p. 19; Monumenta Henricina 1960, vol. 1, doc. 12, pp. 26–27).

Through the second bull, "*Iustis petentium desideriis*" of 8 August 1196, in the context of the complex and long dispute over the churches of Pombal, Ega, and Redinha, located south of Coimbra (Marques 1987, pp. 349–66), Celestine III confirmed the privileges of the Knights Templar and declared that those churches were "*Sarracenorum manibus liberantes*", and were directly subject to Rome, persisting in the idea of liberating the territory from the Saracens (Monumenta Henricina 1960, vol. 1, doc. 15, pp. 31–32).

At the time of the conquest of Alcácer do Sal, the way in which the Holy See approached these issues reveals some relevant specificities. This can be explained by the proclamation of the Fifth Crusade in 1215 and the whole atmosphere that marked the Fourth Lateran Council. By the bull "*Ad liberandam Terram Sanctam*", dated 14 December 1215, Innocent III urged Christendom to defend the Holy Land and persisted in the idea of its liberation. However, the words he chose to do so point to the aggravation of the discourse. This diploma contains expressions such as "*ad liberandam Terram Sanctam de manibus impiorum*", next to others already usual among these written sources: "*Sarracenis*" or "*terras Sarracenorum*". Two new topics then arose: the mention of the "*subsidium Terre Sancte*" and the designation of those who considered themselves "*crucesignatos*" (Bulário Português 1989, doc. 207, pp. 367–71). Within the corpus of documents selected for this research, this emblematic expression—"*crucesignatos*"—appears for the first time in 1215 in a context of pontifical documentation, related in particular to the Iberian Peninsula. Until 1215, the word crusader was not used. Goñi Gatzambide pointed to the year 1300 as the first time the word crusade was used in Iberia (Goñi Gatzambide 1958, p. 232). This demonstrates the difference in occurrence and usage between the concept of crusade and that of crusader. The former is virtually absent from medieval texts whereas the latter is employed in a regular fashion (Maier 2021; Weber 2021). To refer to the movement towards the holy places and the people associated to it, a variety of expressions were used, generally relating to the concept of journey, pilgrimage and an army wearing the cross. In the beginning, the movement was envisaged as a spiritual experience that required an internal vow by the participant. Thus, a campaign like this was seen as a mission, enhanced by the pilgrimage. People who participated in those missions did not have the same perception as the authors of the canonical and juridical standards of the crusade itself (Tyerman 1995, pp. 555–56). Riley-Smith tried to identify the linguistic expressions prior to and announcing the First Crusade, highlighting the way in which Urban II valued the growing importance of the Cross as a symbol of Christian identity and the reference made by him, in 1093, to Muslims as "enemies of the cross" (Riley-Smith 2003). In the literary narratives by northern Europe's "*peregrini*" that were dedicated to the three territorial conquests in Portugal, this concept would only appear two years later, in 1217, as pointed out above.

In a 1984 paper, M. Markowski attributes the official consecration of the concept "*crucesignatus*" to Innocent III, precisely in the context of the Fourth Lateran Council, although the pope had already used the word in 1199. The papal appropriation of this word was also due to his contact with Gerald of Wales, an Anglo-Saxon intellectual who already used the word "*crucesignatus*" in the chronicles he wrote (Markowski 1984, pp. 157–65). However, C. de Ayala Martínez writes that it was only with Honorius III that the term "*crucesignatus*" was applied to crusaders, although he hypothesises that it had been known in Castilian and

Leonese territories since the mid twelfth century, citing the case of "*Petro Cruzat*", who was so characterised in 1167 (Ayala Martínez 2004). For B. Weber, the first known allusion to the term *crusade* in the context of the war against the Saracens, and in particular the Christian victory at the Battle of Navas of Tolosa, can be found in a 1212 letter of donation of two vineyards to the Hospital of Cizur, in Navarre, in line with the terminology used in southern France (Weber 2016, p. 225).

Following the Fourth Lateran Council, Honorius III's discourse became more severe and acquired an ideological tone. According to C. de Ayala Martínez, the papal chancery had been showing signs of catastrophism and radicalisation of terminology since the end of the 1210s (Ayala Martínez 2021, pp. 41–74). In the documentation addressed to Portugal, if some expressions from the old model in "*Manifestis probatum*" (1179) continued to be reproduced on the one hand, then, on the other, significant innovations were introduced, as we will highlight below. In the bull "*Vestris piis postulationibus*" (18 January 1217), in which the pope granted the master and the friars of the Order of the Temple the right to build settlements, castles, churches and cemeteries in the lands conquered from the Saracens, some expressions from the 1179 bull were repeated, such as "*populus christianus a sarracenorum eripuerint manibus*" (Monumenta Henricina 1960, vol. 1, doc. 20, pp. 40–41). The same applies to the pontifical diploma confirming "*Manifestis probatum*", on 11 January 1218 (Monumenta Henricina 1960, vol. 1, doc. 27, pp. 50–51).

At the time of the conquest of Alcácer, in his letters "*Intellecta ex vestris litteris*", Honorius III informed the bishops of Lisbon and Évora, the Master of the Order of the Temple in Spain, the Prior of the Hospitallers in Portugal, and the Commander of the Order of Santiago in Palmela that he did not wish to divert the crusaders from the Holy Land. Instead, the pope urged them to persuade the "*crucesignati*" to remain in the Peninsula, in order to "*expugnandum inimicos nominis christiani*", in exchange for the granting of a plenary indulgence on equal terms with those who served in the holy places. The pope also appealed for their participation in repairing and defending the castle of Alcácer (Monumenta Henricina 1960, vol. 1, doc. 28, pp. 52–54).

Honorius III continued to defend the cause of the crusade. In the pontifical brief "*Cum in generali concilio*" of 30 January 1218, he recommended to the Archbishop of Toledo that the kings and princes of Hispania should make peace, or at least a truce of four years, so that the faithful could freely repress the so-called *infidels*: "*ad insolentiam infidelium reprimandam*" (Monumenta Henricina 1960, vol. 1, doc. 30, pp. 55–56). The idea of repressing and oppressing the *infidels* constitutes another discursive resource that pushes towards an action that is intended to be more generalised. In the immediate aftermath, the papal briefs "*Certum est*" were sent to the King of Leon, urging him to comply with the already mentioned recommendation to the Archbishop of Toledo regarding the Saracens. The action was aimed "*ad exterminandos sarracenos ipsos de Yspanie finibus et christianorum terminos dilatandos*" (Monumenta Henricina 1960, vol. 1, doc. 31, p. 56), that is, the extermination of the Saracens and the expansion of the frontier of Christendom. As in no other written record examined here, the latter appeals to discursive resources marking the radicalisation of the pontificate's position and accentuating the opposition between Christians and Muslims.

### 3.3. Royal Written Records

After the conquest of Santarém, King Afonso Henriques donated the ecclesiastical estates in this town to the Knights Templar in fulfilment of a vow made in April 1147: "*propositum feci in corde meo et votum vovi quod si Deus sua misericordia illud mihi attribueret omne ecclesiasticum darem Deo et militibus fratribus Templi Salomonis*". Evoking the motivation to prepare for the conquest of Lisbon, he wrote: "*Sed si forte evenerit ut in aliquo tempore michi Deus sua pietate daret illam civitatem que dicitur Ulixbona*" (Documentos Medievais Portugueses. Documentos Régios 1958, vol. 1, t. 1, doc. 221, p. 272; Monumenta Henricina 1960, vol. 1, doc. 2, pp. 3–4).

Amongst the preparations for the conquest of Lisbon in June 1147, King Afonso Henriques made a pact with the so-called "*francos*", "*franci*", "*francorum*" (Documentos Medievais Portugueses. Documentos Régios 1958, vol. 1, t. 1, doc. 223, p. 274), without indicating any religious motivations. The same can be observed in the king's donation, in 1148, to Guilherme de Cornibus of Atouguia da Baleia (Peniche), in acknowledgement of his help and good services "*in captione de Ulixbona*" (Documentos Medievais Portugueses. Documentos Régios 1958, vol. 1, t. 1, doc. 225, p. 276). As part of the reorganisation of the city after the conquest, on 8 December 1149 the king donated 32 houses and other assets formerly belonging to the Mosque to the See of Lisbon, i.e., "*quas omnes misquite in tempore sarracenorum habuerunt*" (Documentos Medievais Portugueses. Documentos Régios 1958, vol. 1, t. 1, doc. 232, p. 284).

Indeed, in the context of the preparation for the conquest of Silves, King Sancho I confirmed the possession of Porto to the city's bishop and chapter, exempted the inhabitants from royal services and granted several privileges to the clergy of the whole kingdom. Namely, they were dispensed from participating in the royal army "*nisi contra sarracenos si intrauerint in terram nostram*" (Documentos de D. Sancho I (1174–1211) 1979, doc. 39, pp. 62–63). Thus, the king signalled his commitment to organising the conquest of Silves, replicating what had been his expectations during the preceding conquest of Lisbon, as regards obtaining the collaboration of the northern bishop in mobilising resources for the conquest.

After the victory in Silves, King Sancho I arranged the distribution of assets to gratify those who had supported the military operation and ensured the town's governance. Concretely, in December 1189, the king donated the castle of Alvor, located in the territory of Silves "*in terra Sarracenorum*", to the monastery of Santa Cruz de Coimbra (Documentos de D. Sancho I (1174–1211) 1979, doc. 41, pp. 64–65). Around the same time, he donated the town of Mafra to Nicolau, Bishop of Silves, as well as some privileges and properties misappropriated by the Templars. The bishop also received the first revenues from the Templars, the Hospitallers, and other orders, as well as the right to collect tithes. The king also forbade them from building churches in the diocese of Silves (Documentos de D. Sancho I (1174–1211) 1979, doc. 42, pp. 66–67). This document is quite significant in regard to military orders, commonly associated with this type of battle, considering the context in which they were placed. Drawn up after the Christian victory over the Muslims, and thanking the Normans for this success, the king used this document to limit the interests of the Jerusalemite orders (the Temple and the Hospital) in favour of those of the bishop and, at the same time, to belittle the collaboration of the Normans in the conquest. On a different note, on 27 July 1190, among the beneficiaries of a donation, King Sancho I included the monastery of Grijó, to which he donated the "*fossadeiras*" (the amount paid by those who did not take part in the war against the enemies) "*pro remissione peccatorum nostrorum et pro amore uassali nostri domni Aluari Martini qui in obsequio Dei et nostro contra inimicos crucis Christi apud Siluim interfectus est a Sarracenis*" (Documentos de D. Sancho I (1174–1211) 1979, doc. 44, pp. 68–69). Santa Cruz de Coimbra and Grijó were Augustinian monasteries, and they both had high religious and political prestige at the time.

In royal documents addressed to the Military Orders in the years that followed, there are no references to motivations related to the war against the Saracens, nor to any interventions from the Orders in these matters. This was the case on 1 January 1193, when King Sancho I donated to the Order of Santiago some houses in Santarém located where King Afonso Henriques had entered the town when conquering it (Documentos de D. Sancho I (1174–1211) 1979, doc. 64, p. 100–1). The same can be said of the donation of the castle of Mafra and its border to Gonçalo Viegas, master of the Order of Évora (later the Order of Avis) (Documentos de D. Sancho I (1174–1211) 1979, doc. 65, pp. 101–2). This omission is even more significant given that, around 1190–1191, Portugal was the target of an intense Almohad wave of attacks, which caused the border to shrink considerably into the north. In fact, it was following this Almohad counterattack that, on 13 June 1194, King Sancho I donated the Guidimtesta estate to the Order of the Hospital, requiring the friar

knights to build a castle, which he called Belver (Documentos de D. Sancho I (1174–1211) 1979, doc. 73, pp. 112–13), without, however, making any reference to warfare associated with the crusade.

In short, King Sancho I did not seem to acknowledge the contribution of the military orders and the northern soldiery to the conquest of Silves in 1189. At first, coinciding with the conquest itself, the king was hostile towards them, and in the 1190s, when he made donations to them, he never mentioned the climate of war to which they were traditionally committed. In contrast, when he made donations to the monasteries of Santa Cruz de Coimbra and Grijó, the king included references to this type of warfare. This decision may have several explanations. First, there was an unfavourable international situation for the Templars and Hospitallers in the wake of their defeat at the Battle of Hattin in 1187, which resulted in them losing control over Jerusalem. The following year, Saladin continued his conquests against the eastern Frankish states. In the Iberian Peninsula, too, Christian losses were mounting, and this may have fed a wave of mistrust towards the martial capability of the military orders. The Almohad wave in the early 1190s led to Abu Yakub al-Mansur's advance further north reaching the castle of Tomar, giving rise to the loss of Alcácer do Sal on 10 June 1191, followed by the surrender of Palmela and Almada, and the Muslim retaking of Silves on 10 July 1191 (Branco 2006, p. 142, map with 1190–1191 campaigns; Cunha 2021, p. 50). In Castile, there was a heavy Christian defeat by the Muslims at the Battle of Alarcos in 1195. In 1212, this defeat would be redeemed by the Christian victory at the Battle of Navas de Tolosa, led by King Alfonso VIII of Castile, during the reign of King Afonso II of Portugal (1211–1223).

The new Portuguese monarch, Afonso II, developed a policy of strengthening royal power which led to long conflicts with the clergy, with the king being excommunicated by the pope (Vilar 2005). Between the end of July and 18 October 1217, the campaign of Alcácer do Sal took place in this context, precisely in a former estate of the Order of Santiago. This was the third campaign to be achieved thanks to the intervention of a northern European fleet heading for the Latin East to join the Fifth Crusade. Once the victory was achieved, the process of organising the town's Christian administration began under royal initiative. In 1218, Alcácer was granted the royal charter, which included a clause equating Christians, Jews, and Moors: "*ut quicunque pignoraverit mercatores vel viatores chritianos, iudeos, sive mauros*" (Portugaliae Monumenta Historica a saeculo octavo post Christum usque ad quintumdecimum iussu Academiae Scientiarum Olisiponensis edita. Leges et consuetudines 1864, vol. 1, fasc. IV, p. 581) in a very particular situation.

Among the documents granted immediately after the events of Alcácer, no other significant discursive resources were identified, although the struggle for control of the territory remained active.

## 4. Conclusions

Identifying the discursive resources used at the time to signify the crusade involves an exercise of precaution, which is incompatible with generalisations. As A. Papayianni has observed (Papayianni 2016, pp. 278–80 and p. 284), the diversity of terms used is broad, both in the West and in the Byzantine East. The assimilation of these terms and the fixation on the word crusade have a long history. This complexity results from the newness of that historical process, which was yet to fix a word that synthesised, conceptualised, and legitimised the crusade and took into account its geographic context. Means had to be found to create cohesion between the Holy Land and the westernmost boundary of Iberia. The word crusade corresponds to a category common to several realities, emphasising the similarities between them and allowing their dissemination and assimilation by society in general (Weber 2016, pp. 221–33). In short, the concept of crusade emerged gradually, according to the historical evolution of the situation to which it sought to give meaning, and so it was not static. At the same time, the militarisation of society and territory was evolving, a process that had a profound influence on the acceptability of the crusade.

The long time it took to fix and crystallise the word crusade, as an instrument of the consecration of its conceptual newness and of its standardisation and generalised acceptance, may have some relation to the fact that we are dealing with a historical period in which a rapid succession of rule-makers (and opinion makers) is observed. In the 70 years between the conquest of Lisbon (1147) and the conquest of Alcácer do Sal (1217), several chronicles were written by some northern men, eleven popes ruled Christendom (Eugenius III, Anastasius IV, Adrian IV, Alexander III, Lucius III, Urban III, Gregory VIII, Clement III, Celestine III, Innocent III, Honorius III) and three monarchs reigned in Portugal (Afonso Henriques, Sancho I, Afonso II). They had distinct world views and, therefore, their goals were not necessarily concurrent. This diversity contributed to fixing terminological expressions and concepts associated with that same worldview. Each of the three main written narratives represents literary and theological traditions which sit outside Portugal, and each of them reflects distinct experiences in Portuguese territory. Unfortunately, there are no written sources written in Portugal to complete this point of view. Although the western part of the Iberian Peninsula was considered a reconquest territory, somehow similar to the Holy Land, and where the military orders had an intensive role, the word crusade was not used in advance, and it had no particular clarification. This can be justified since Portuguese authors did not write the selected texts.

Historiography is often compelled by ideological underlying intentions. Sometimes, it reveals a tendency to repeat ideas without critically discussing them. More recent scientific literature has sought to deconstruct preconceived notions and to value new research approaches toward the specificity of each case based on a continuous re-examination of written sources.

This paper has analysed three groups of documents: the crusaders accounts of the conquests of Lisbon (1147), Silves (1189) and Alcácer do Sal (1217); the records of the royal chancery; and the pontifical documents, with the latter two written shortly after the military events. Two fundamental questions underlie this approach: were the contemporary crusader narratives, the pontifical discursive resources, and the Portuguese royal discourses consistent with each other? Was there any diachronic evolution in this matter between 1147 and 1217?

From the perspective of the conquest narratives, those who took part in the military actions were seen as "*peregrini*", which, according to the literature, were not only those who undertook a religious journey, but also those who had left their homeland to travel to foreign lands. The word expresses the reason which moved them, from the moment they were in Iberian territory, having left northern Europe for the Latin Orient. However, due to the political and military situation in the west of the peninsula, they were encouraged to participate in those conquest campaigns. Convincing them was not easy, and only the royal promise of access to looting seemed to motivate them. The written records of the papacy and the monarchy do not designate them as pilgrims, giving preference instead to the word "*francorum*" and, thus, highlighting the origin of several of them. Both the papacy and the crown called for the participation of these people in more polarised terms, encouraging confrontation between the Christian and Muslim worlds. Both intended to use the potential of these northern human resources to enlarge the armies that would make possible the expansion of Christendom in general and of the kingdom of Portugal in particular. The papacy's discursive resources were more radical and incorporated religious and even moral motivations, which is unsurprising. Besides the ideological dimension in the papacy's approach to this type of warfare, perhaps the unfamiliarity with the real situation in Iberia justified a magnification of the discourse, with the aim of mobilising people for these actions to obtain the expected results.

Taking into consideration these conclusions, in future research we intend to study the impact of all these written records on the military orders, assessing to what extent such records were a source of inspiration and how they may have contributed to an education of the friars of these institutions with regards to the late medieval conception of crusade.

**Author Contributions:** Conceptualization, P.P.C. and J.L.; methodology, P.P.C. and J.L.; software, P.P.C. and J.L.; validation, P.P.C. and J.L.; formal analysis, P.P.C. and J.L.; investigation, P.P.C. and J.L.; resources, P.P.C. and J.L.; data curation, P.P.C. and J.L.; writing—original draft preparation, P.P.C. and J.L.; writing—review and editing, P.P.C. and J.L.; visualization, P.P.C. and J.L..; supervision, P.P.C. and J.L.; project administration, P.P.C. and J.L.; funding acquisition, P.P.C. and J.L.. All authors have read and agreed to the published version of the manuscript.

**Funding:** This paper is financed by National Funds through the FCT—Foundation for Science and Technology, under the project UIDB/04059/2020.

**Informed Consent Statement:** Not applicable.

**Conflicts of Interest:** The authors declare no conflict of interest.

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
