# Peer review of "Crusade: The Arising of a Concept Based on Portuguese Written Records of Three Military Campaigns (1147–1217)"

_religions, doi:10.3390/rel14020244_

Round 1
Reviewer 1 Report
The stated aim of this article is an ‘analysis [...] based on [...] written records in order to clarify when the word crusade/crusader appeared in Portugal, to assess to what extent the war of conquest in Portuguese territory followed the example of the holy war and to evaluate the commitment of the Crown and the Holy See in this complex process” (ll.27-31). The author examines narrative accounts of three military campaigns for the conquest of Lisbon (1147), Silves (1189) and Alcácer do Sal (1217), as well as royal and papal documents relating to these events. The author comes to the conclusion that generally “the concept of crusade emerged gradually, according to the historical evolution of the situation to which it sought to give meaning” (ll.460-1) and that in the Portuguese context “[t]he written records of the papacy and the monarchy do not designate them [i.e. crusaders] as pilgrims, giving preference to the word “francorum”, thus highlighting the origin of several of them. Both the papacy and the crown called for the participation of these people in more polarised terms, encouraging confrontation between the Christian and Muslim worlds” (ll.494-8). Just as these results are at best tenuous answers to the questions posed at the beginning of the article, so the overall argument is deeply flawed on several levels.
The main shortcoming of this article is the fact that the author is not au courant with, and does not use, the basic and most recent research on the topics studied. In addition to this, the purported focus of the investigation is not sufficiently analysed and thought through. There is, in the medieval record, a basic difference in occurrence and usage between the concept of crusade and that of crusader. The former is virtually absent from medieval texts whereas the latter is employed in a regular fashion. This dichotomy as well as the meaning and use of the two concepts have been analysed and contextualised in a variety of recent studies which the author does not seem to be aware of. For this see: Trotter, David A., Medieval French Literature and the Crusades (1100-1300), Geneva, 1988; Hehl, Ernst-Dieter, 'Was ist eigentlich ein Kreuzzug?', Historische Zeitschrift 259, 1994, pp. 297-336; Tyerman, Christopher, 'Were There Any Crusades in the Twelfth Century?', English Historical Review, 110, 1995, pp. 553-77; Tyerman, Christopher, The Invention of the Crusades. Basingstoke and London, 1998; Weber-Vivat, Bejamin, 'Nouveau mot ou nouvelle réalité? Le terme cruciata et son utilisation dans les textes pontificaux', in Michel Balard, ed. La Papauté et les croisades / The Papacy and the Crusades. Actes du VIIe Congrès de la Society for the Study of the Crusades and the Latin East/ Proceedings of the VIIth Conference of the Society for the Study of the Crusades and the Latin East (Crusades – Subsidia, 3). Farnham, 2011, pp. 11-26; Cosgrove, Walker Reid, 'Crucesignatus: A Refinement or Merely One More Term among Many?', in Thomas F. Madden et alii, eds. Crusades – Medieval Worlds in Conflict. Farnham, 2010, pp. 95-107. Maier, Christoph T., 'Pope Innocent III and the crusades revisited', in P. Ingesman (ed.), Religion as an Agent of Change. Crusades – Reformation – Piety, Leiden, 2016, pp. 55-74; Maier, Christoph T., 'When was the First History of the Crusades Written?', The Crusades. History and Memory. Proceedings of the Ninth Conference of the Society for the Study of the Crusade and the Latin East, Odense, 27 June – 1 July 2016. Volume 2, Turnhout, 2021, pp. 13-28; Weber, Benjamin, ‘When and where did the word ‘crusade’ appear in the Middle Ages? And Why?,’ in The Crusades. History and Memory. Proceedings of the Ninth Conference of the Society for the Study of the Crusade and the Latin East, Odense, 27 June – 1 July 2016. Volume 2, Turnhout, 2021, pp. 199-220. Weber, Benjamin, ‘Conceptualizing the Crusade in Outremer: Uses and Purposes of the Word “Crusade” in the Old French Continuation of William of Tyre’, Crusades 20, 2021, 151-164.
In terms of the author’s argument as it stands, the main problem, apart from the missing distinction between terms for crusade and terms for crusader, is the lack of an apparent methodology. Throughout the argument it is not clear what is being investigated. The article basically consists of a listing of short passages from various narrative accounts and administrative documents in which crusaders fighting in the three Portuguese campaigns as well as their Muslim enemies are described in varying terms. But there is no real attempt at working out why some of the terminology appearing in these passages can be deemed as reflecting the language of crusade. This is in part caused by the lack of a discussion of the relevant literature quoted above and the failure to determine what can be considered to be crusade terminology in other crusading context in texts from the same time. It is thus not surprising that the article produces no tangible and clear results.
Overall, the author does not employ sufficiently analytical language. The argument is often vague and at times confusing. This has in part to do with the fact that, quite obviously, the author is not a native English speaker. Although basic grammar and sentence structure are largely correct, there are some serious problems on the lexical and stylistic levels which contribute to the confusions inherent in the argument. E.g. the use of the terms ‘resource’, ‘foreign’, etc. need to be refined; other terms are clearly mistranslations, e.g. ‘Northmen’, ‘Teutonic pilgrim’, ‘Compostela [being] an apostolic see’, etc. There are also some interpretations which are not tenable as they stand, e.g. the interpretation of looting after the conquest of Lisbon as ‘economic intentions’ on the side of the crusaders (ll.149-51): looting was part and parcel of any medieval war and cannot be taken as an indication of the motivation behind a military campaign, or: the idea that only crusaders but not ordinary pilgrims took vows (ll.210-2) is equally not true, or: a document of 1217 cannot possibly have been addressed to Pope Urban III (l.233) etc. The bibliography suggests that the author is not widely read and primarily uses Portuguese texts. This leads to omissions like Carl Erdmann’s 1930 article on crusading in Portugal, Wendell David’s edition of De Expugnatione Lyxobonensis and works by Marcus Bull, Hugh Reilly, Hugh Kennedy, Alexander Bronisch, David Lomax, James O’Callaghan, Robert Burns, Nikolas Jaspert and others which treat important aspects of the reconquista/crusade on the Iberian Peninsula. There are also mistakes: Riley-Smith’s The First Crusade and the Idea of Crusading was published in 1986 (2003 is simply a reprint).
Author Response
REVIEWER 1:
The review report, after summing up the main points of the work, argues that the “results are at best tenuous answers to the questions posed at the beginning of the article”. The purpose is indeed a lexicography approach based on the selected words used in Portuguese and pontifical documents for describing the three conquests.
This is an original article. Concerning the historiography, the goal is not to do the state of the art of the topic. The paper was prepared to be published at a specialized scientific journal, and for a special issue (The Crusades from a Historical Perspective: Communications, Culture, and Religion), and neither to a book nor to an extensive essay.
Although there is a difference in occurrence and usage between the concept of crusade and that of crusader, the selected Portuguese written records for the analysis put in evidence the absent of both. Only doing this type of lexicography approach is possible to improve the knowledge without repeating the ideas from the former historiographic outputs.
Of course, we are aware of the recent studies, and we are scientific updated. We add some titles to the new version of the paper.
The methodology used consists in the reading of the written records, and on the selection of the short passages that show how the military conquests of Lisboa, Silves, and Alcácer deal within the contribution of the Northern fleets committed with the crusade movements. The text is based on analytical language.
The authors are not native English speaker, but the translation of the text was made and revised by an expert. Some amends are done in the new version. As we all know, the scientific knowledge benefits from the contribution of the international community, which implies to accept distinct ways of expression.

Reviewer 2 Report
The main interest of this paper concerns the lexicography: what words are used in Portuguese and pontifical documents for the conquest of three main Portuguese cities from the Sarracens, and when is appearing the word crusade in these documents.
Author Response
REVIEWER 2:
The review report points out the main goal of the paper. Nothing to add.

Reviewer 3 Report
This article looks to argue that, through analysis of the written accounts of three campaigns, Lisbon, Silves, and Alcacer do Sal, that there is no clear sense of a ‘crusade’ or ‘crusaders’ until the 13th century (corresponding with the rise of the term crucesignati elsewhere). The concept of the article is interesting, and in many ways it achieves its, but I am left wondering quite what the analytical purpose is, in terms of filling a gap in the scholarship. Indeed, on the whole I don’t think it’s quite ready for publication and needs a much better sense of why it is actually important for scholarship that it exists.
For a start, the introduction does not do all that much to set up the existing field, and though it recognises some up to date stuff, there are some serious omissions. These include William Purkis’ on crusader spirituality; Jonathan Phillips on holy war in the De Expugnatione; recent work by Jonathan Wilson on this text; Luis Garcia Guijarro on the siege of Lisbon/Iberian campaigns as crusading (seen in Roche et al’s recent edited collection on the Second Crusade); and maybe Simon John’s Journal of Ecclesiastical History article that discusses the foundation of the kingdoms of Portugal and Jerusalem. There is a need, therefore, to better situate this piece against existing scholarship to better show why it is necessary.
I wonder, also, why the author has chosen to prioritise the word crusade over the institutional aspects of crusading. This has sparked some debate, of course, but the recent work of Benjamin Weber (only one of which is listed here), has made the astute point that the word came into use not to create the idea of crusade, but rather when it was that the idea became challenged, and so definition was required. It was thus a move to clarify what was already well-known. In part, this sits very neatly with the work of Riley-Smith et al., but also I think the more literary driven works of people like Marcus Bull, Stephen Spencer, and Katherine Allen Smith, that has sought to consider the broader development of the ideal/institution. I think the author needs to, again, be more attuned to this and to make some allusion to this. Moreover, even if we do just stick to terminology, I think the work on the use of the word ‘pilgrim’ needs some work, as the author recognises (using Riley-Smith) that this is a recognised term for crusaders in Latin narratives but then discusses the notion of the word meaning outsiders. A bit more direct comparative work with crusade texts outside of Iberia here is required to draw out this point. Keeping with this, I think the analytical reasoning behind the discussion of terms/descriptions used for Muslims needs better discussion. It’s not entirely clear why this is seen as part of the definition of a crusade here – so it needs proving that this would actually be something that would indicate crusading. To my mind, neither of the First or Second Crusades, though of course primarily launched against Muslims, are defined by this, because they’re seen more as an internal vow by the participant.
Likewise, the author does not appear to know, or at least does not discuss, the debated nature of the De expugnatione’s form and author, like Jonathan Wilson’s recent work, which actively challenges the identification of Ralph and argues for a Norman settler, Robert. There is also a new edition of the conquest of Silves text (by Dana Cushing).
Linked to this issue of source work is the fact that, while the author seems to want to discuss how the crusade was seen in Portugal, they are working with narratives, for the most part, not seemingly written by a local. Each of the three main written narratives represent literary and theological traditions which sit outside of Portugal. What value do they have, then, in such a discussion? Some discussion of this is required to better set up the analysis.
When discussing Compostella in page 5, the author might want to dig a little more into wondering how those outside of Iberia might know about its privileges. There’s plenty of word on marriage alliances between Spain and France (and England), but also it’s a pilgrim destination and we are talking about a period when there had been two recent Lateran Councils (3 and 4) which would have brought churchmen from across Christendom together.
I would also suggest, should you hope to appeal to non-specialist/undergraduate readers, that all Latin quotes are also translated into English.
Some specific points:
P. 1, l. 37 – what is this ‘bull of crusade’? It surely warrants some form of active description
p. 1, l. 42 – maybe the author could cite some of this military orders scholarship?
p. 4, l. 185 – the correct word is Jerusalemite
p. 4, l. 185-6 – issue with the comma across the line break
p. 4, l. 188 – the correct word is Franks
p. 6, l. 306 – it simply cannot be said that Gerald of Wales was an Anglo-Saxon. Anglo-Norman is fine.
p. 8, ll. 392-3 – Jerusalemite again
p. 9, l. 425 – Frankish, not French
p. 9, ll. 427-430 – Sentence doesn’t make sense. Changing ‘gave’ to ‘giving’ should sort it.
p. 10, l. 469 – What does the author mean by Nordic chronicle?
Author Response
REVIEWER 3:
This is an original article. The goal of the paper is not to do the state of the art of the topic. The paper was prepared to be published at a specialized scientific journal, and for a special issue (The Crusades from a Historical Perspective: Communications, Culture, and Religion), and neither to a book nor to an extensive essay. We are scientific updated, and some titles were added to the new version of the paper. The aim is to do a lexicographic approach. We have chosen to prioritize the word crusade over the institutional aspects of crusading because some authors insist to use this word to report the Iberian situation.
As this paper was prepared to a specialized scientific journal, and for a special issue (The Crusades from a Historical Perspective: Communications, Culture, and Religion), the Latin quotes were not translated into English.
The authors are not native English speaker, but the translation of the text was made and revised by an expert. Some amends are done in the new version.

Reviewer 4 Report
It would be useful to compare the author's findings with similar phenomena in the Latin East,
There are no references to Christopher Tyerman's works throughout the article.
In line 95, the author writes: "Its author, probably a Teutonic pilgrim, wrote at the end of the twelfth century." The term "Teutonic" demands clarification: Does it mean someone of German origin? Member of the Teutonic order?
Author Response
REVIEWER 4:
The comparative approach with similar phenomena in the Latin East is out of the goals of this paper, since the main purpose is to trace the terminology used in the selected Portuguese documents and to identify how the historical conquests of Lisbon (1147), Silves (1189), and Alcácer do Sal (1217) were referred to, concerning a lexicography approach, are the two main goals of this paper.
Christopher Tyerman's works were added.
In line 95, the term "Teutonic" means someone of German origin.

Round 2
Reviewer 1 Report
This revision is not answering the criticism and suggestions of the previous review. The only substantial changes made are some sentences added for referencing recent research which was missing in the original text. But there is no discussion of the findings presented in these publications nor does the author make an effort to work these findings into the argument of the present article. The lack of methodology, of precise terminology, of a clear research question and of conclusive results has not been addressed in the revised version.
Author Response
We appreciate the comments made at the first round, and according to them, we carried out important changes.
There is a misunderstanding. A deep discussion and a review of the methodology would give rise to a new article. The one we submitted is based on a deep analysis of the documents, and precise terminology.
Concerning the item “Are all the cited references relevant to the research?”, the reviewer, in the first round had chosen the option “yes” and now, after major additions concerning cited references, he/she chose “can be improved”.
We have again made changes to the text to clarify the issues suggested by the reviewer.

Reviewer 3 Report
The author has addressed some of the issues raised in the peer review report, albeit not all. Some are rather clumsily attempted (see the rather odd inclusion of a comment on the non-Portuguese nature of the three narratives in the conclusion, which does not really address the concern that these texts do not reflect consistent/similar traditions). Still, there are some improvements and a better engagement with some of the scholarship. Some of the new sentences are a little clumsy in their English expression as well and the introduced Tyerman quotes - presumably brought in because of his work on the 'invention of crusading' in the 12th century - are a little odd. It seems the author is not really engaging with the ideas of Tyerman, just finding quotes about the events in question. Maybe they might want to rethink this approach.
Author Response
We appreciate the comments made at the first round, and according to them, we carried out important changes. We have again made changes to the text to clarify the comments of round 2.
